# Super-resolving normalising flows for lattice field theories

**Marc Bauer[1], Renzo Kapust[1*], Jan Martin Pawlowski[1,2] and Finn Leon Temmen[1]**

**1** Institute for Theoretical Physics, Universität Heidelberg,
Philosophenweg 16, D-69120, Germany
**2** ExtreMe Matter Institute EMMI, GSI, Planckstr. 1, D-64291 Darmstadt, Germany

★ kapust@thphys.uni-heidelberg.de

## Abstract

We propose a renormalisation group inspired normalising flow that combines benefits from traditional Markov chain Monte Carlo methods and standard normalising flows to sample lattice field theories. Specifically, we use samples from a coarse lattice field theory and learn a stochastic map to the targeted fine theory. The devised architecture allows for systematic improvements and efficient sampling on lattices as large as $128 \times 128$ in all phases when only having sampling access on a $4 \times 4$ lattice. This paves the way for reaping the benefits of traditional MCMC methods on coarse lattices while using normalising flows to learn transformations towards finer grids, aligning nicely with the intuition of super-resolution tasks. Moreover, by optimising the base distribution, this approach allows for further structural improvements besides increasing the expressivity of the model.

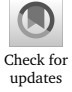

# 1  Introduction

Lattice field theory constitutes one of the primary methods for solving quantum field theories non-perturbatively. For sampling from the respective statistical integrals, traditional Markov chain Monte Carlo (MCMC) methods have been effectively employed, providing valuable insights into the physics of various many-body systems.

The limitations of traditional MCMC methods, such as the need for individual simulations for different points in the parameter space and critical slowing down near a second-order phase transition, have prompted the exploration of new approaches for sampling Boltzmann distributions.

Moreover, a crucial aspect in lattice field theory is the continuum limit which requires an access to finer and finer lattices for an extrapolation toward the continuum theory of interest. As the continuum limit constitutes a second-order phase transition itself and is inherently computationally expensive this further motivates the search for more efficient algorithms.

With the advent of machine learning techniques in physics, normalising flows have emerged as a prominent candidate to alleviate or transform the challenges of traditional MCMC sampling approaches. While delivering impressive results, normalising flows come with their own issues, including unfavourable volume scaling of the complexity of the model and mode collapse in multimodal distributions.

In this work, we combine the benefits of traditional MCMC methods and normalising flows rather than replacing one by the other. We do so by noting that normalising flows have been used in the machine learning community to tackle super-resolution tasks [1–3] effectively. There, one wants to obtain a high-resolution image from a low-resolution one. In the context of lattice field theory, this translates to learning an inverse renormalisation group step towards the continuum limit from a given coarse lattice.

Recently, first steps to investigate multiscale normalising flows for a lattice field theory as well as learning (inverse) renormalisation group transformations have been made [4–13]. In this paper, we propose a normalising flow architecture that connects a coarse and fine lattice while incorporating intuitions of the renormalisation group. The coarse lattice hereby stems from traditional MCMC algorithms while the stochastic map to the finer lattice is learned by the flow. Importantly, the actions on the coarse and fine lattice are both given by structurally the same (interacting) lattice action. While the couplings on the fine lattice are kept fixed, we show how to optimize the coarse couplings in order to achieve the best sampling quality on the fine lattice.

In practice, we demonstrate the sampling of field theories on lattices as large as $128 \times 128$ in the symmetric and broken phase when running an efficient MCMC method on a $4 \times 4$ system. Moreover, by starting on finer grids or further adapting the base distribution, the methods presented here allow for a systematically improvable architecture beyond making the learned transformations between the grids more expressive.

The paper is structured as follows: In Section 2, we introduce the lattice field theory setup and the $\phi^4$-theory as a canonical example. In Section 3, we introduce square and rectangular normalising flows and discuss their application to lattice field theory. This enables us to present the renormalisation group inspired normalising flows in Section 4 and to discuss the used optimisation criterion as well as the optimisation of the used couplings in Section 5. We then showcase the newly modulated architecture in Section 6, where we fix the distribution on the fine lattice and then find the best transformation and coarse lattice distribution for this sampling task. We conclude in Section 7.

## 2 Scalar lattice field theory

In this work we only consider a scalar field theory, but we emphasise that our general approach is not restricted to it. For the scalar theory the Euclidean path integral is given by

$$\mathcal{Z} = \int \mathcal{D}\phi \, e^{-\mathcal{S}[\phi]}, \tag{1}$$

where the action $\mathcal{S}[\phi]$ is a functional of the continuous field $\phi(x)$. In the following we use the $\phi^4$-theory as an example to illustrate the concepts and methods introduced in this work. Its action is given by

$$\mathcal{S}[\phi] = \int d^d x \left[ \frac{1}{2}\phi(x)\left(-\Delta + M^2\right)\phi(x) + \frac{g}{4!}\phi^4(x) \right], \tag{2}$$

with the mass $M$ and coupling $g$. In the lattice formulation, we discretise the field $\phi(x)$ on a $d$ dimensional hypercubic lattice $\Lambda$ with $L$ lattice sites in each dimension, separated by the lattice distance $a$. We refer to $L$, the number of grid points in each direction, as the *linear lattice size*, which should not be confused with the physical spacetime extent $L\,a$ of the lattice. The discretised lattice action is then given by

$$S(\phi) = \sum_{x \in \Lambda} \left[ (1-2\lambda)\,\phi_x^2 + \lambda\phi_x^4 - 2\kappa \sum_\mu \phi_x \phi_{x+\mu} \right]. \tag{3}$$

Here, we have defined the dimensionless quantities $\kappa$, $\lambda$, and $\phi_x$ via

$$a^{\frac{d-2}{2}}\phi(x) = \sqrt{2\kappa}\,\phi_x, \qquad (aM)^2 = \frac{1-2\lambda}{\kappa} - 2d, \qquad a^{4-d}g = \frac{6\lambda}{\kappa^2}, \tag{4}$$

where $x \in \Lambda$. The continuum theory is retrieved in the limit of $a \to 0$, $L \to \infty$, while keeping the simulated physics fixed. In discretised form, we can think of the path integral as a high-dimensional regular integral

$$Z = \int D\phi \, e^{-S(\phi)}, \qquad D\phi = \left( \prod_{x \in \Lambda} d\phi_x \right). \tag{5}$$

From the statistical physics point of view, the action induces a Boltzmann distribution

$$p(\phi) = \frac{e^{-S(\phi)}}{Z}, \tag{6}$$

on the lattice, and observables $\mathcal{O}(\phi)$ are computed according to

$$\langle \mathcal{O} \rangle = \int D\phi \, \mathcal{O}(\phi) \frac{e^{-S(\phi)}}{Z} \,. \tag{7}$$

Interpreting this expression as a statistical integral enables us to compute the expectation value using Monte Carlo algorithms. Markov chain based approaches, where new samples only depend on its direct predecessor, are the chief method of choice for most lattice computations to date. For a Markov chain Monte Carlo algorithm to be efficient, it must have a low autocorrelation time, which means that the correlation between configurations in the Markov chain should decrease rapidly.

Unfortunately, close to second-order phase transitions, autocorrelation times diverge for any known general MCMC algorithm, as the correlation length diverges. This problem is called critical slowing down [14]. Since the continuum limit constitutes a second-order phase transition of the underlying statistical system, the critical slowing down problem presents itself as a general issue in lattice simulations.

This concludes our quick introduction of the lattice field theory setup.

## 3 Normalising flows

In this section, we discuss normalising flows, which are a general class of machine learning models that allow for efficient sampling and probability density estimation. In the lattice field theory context they were proposed as an interesting alternative sampling method to tackle the critical slowing down problem [15, 16].

Within normalising flows, one tries to find a transformation that maps configurations from a distribution that is easy to sample to configurations that are distributed according to the target distribution of the chosen lattice field theory. Moreover, the flow should do so while efficiently estimating the configurations' likelihood, allowing for exact Monte Carlo approaches.

In the following, we will go into more detail of this sampling approach and introduce square and rectangular flows [17] which will be used to build the proposed architecture of this paper in Section 4.

### 3.1 Sampling with a normalising flow

Assume we want to sample fields $\phi \in \mathbb{R}^{\mathcal{L}^d}$ according to the distribution $p_{\mathcal{L}}$ induced by the action $S_{\mathcal{L}}$ on the lattice $\Lambda_{\mathcal{L}}$ with linear lattice size $\mathcal{L}$. We call this distribution the target distribution.

To utilise a normalising flow for sampling from this distribution, we introduce another field theory with the fields $\varphi \in \mathbb{R}^{L^d}$ that we can sample according to a distribution induced by the action $S_L$ on the lattice $\Lambda_L$. This distribution is called prior or base distribution, and we choose $\mathcal{L} \geq L$.

We now also introduce a map

$$\mathcal{T}_{\theta} : \ \mathbb{R}^{L^d} \to \mathbb{R}^{\mathcal{L}^d} \,, \tag{8}$$

with learnable parameters $\theta$ that connects the fields on $\Lambda_L$ to fields on $\Lambda_{\mathcal{L}}$. In the following, we will mark objects that result from pushing configurations $\varphi$ through $\mathcal{T}_{\theta}$, like the resultant field and its log-likelihood

$$\tilde{\phi} = \mathcal{T}_{\theta}(\varphi), \qquad \log \tilde{p}(\tilde{\phi}), \tag{9}$$

respectively, with an overhead tilde. The normalising flow is expected to estimate both efficiently.

The intuition behind a sampling algorithm based on this setup is the following: Let us assume that we have a base distribution from which we can easily sample i.i.d configurations. Then, starting from these configurations, we optimise the map $\mathcal{T}_\theta$ such that the push forward distribution $\tilde{p}$ approximates the target distribution $p_\mathcal{L}$ well. This allows us to sample from the target distribution by first sampling from the base distribution and then applying the transformation $\mathcal{T}_\theta$. Accordingly, we have transformed the hard sampling problem of the target distribution into the significantly easier one of the base distribution. The costs we have to muster for this transfer are the training costs of the learnable parameters $\theta$.

In many situations, one chooses a standard Gaussian with i.i.d. components as the base distribution. Alternatively, a non–interacting field theory may also be used. In this work, we will consider a plethora of potential base distributions focusing on how they can be constructed and optimised from coarser theories.

Needless to say, the optimisation of the transformation will generally not be perfect. So, an accept-reject step with the acceptance probability

$$\text{acc}(\phi_c, \tilde{\phi}) = \min\left(1, \frac{\tilde{p}(\phi_c)\,p(\tilde{\phi})}{p(\phi_c)\,\tilde{p}(\tilde{\phi})}\right), \tag{10}$$

is necessary to ensure exact sampling in the target theory. Here, $\phi_c$ represents the current configuration in the Markov chain, and $\tilde{\phi}$ represents the configuration proposed by the density-estimating model. Since the total likelihood of each configuration is tracked as it progresses through the flow, we can utilise the likelihood ratio in the acceptance probability [15, 18, 19].

To evaluate the quality of the normalising flow, we follow the literature in considering two further quantities. Firstly, we look at the (reverse) Kullback-Leibler divergence,

$$D_{KL}(\tilde{p}\|p) = \int D\tilde{\phi}\,\tilde{p}(\tilde{\phi})\log\left(\frac{\tilde{p}(\tilde{\phi})}{p(\tilde{\phi})}\right), \tag{11}$$

which provides an (improper) measure of the difference between the push forward distribution $\tilde{p}$ and the target distribution $p$. This quantity is useful because it can be estimated using only samples from the push forward distribution. Secondly, we use the effective sample size (ESS), given by

$$\text{ESS}/N = \frac{1}{\int D\tilde{\phi}\,\tilde{p}(\tilde{\phi})\,w^2(\phi)} \approx \frac{\left(\sum_{i=1}^N w_i\right)^2}{\sum_{i=1}^N w_i^2}, \tag{12}$$

where we have defined $w_i = p(\tilde{\phi}_i)/\tilde{p}(\tilde{\phi}_i)$ for each of the $N$ samples [20]. While a good $\text{ESS}/N$ translates into a good acceptance rate (10), the $\text{ESS}/N$ has the benefit that we must not set up a Markov chain to estimate it.

This concludes the general overview and structure of the sampling algorithm built upon in this work.

Now, we discuss the implications of changing the dimensionality of the field when pushing it through the transformation $\mathcal{T}_\theta$. For that matter, we introduce square and rectangular normalising flows in the following section, and discuss how to compute the (log)-likelihood in these situations.

## 3.2 Square normalising flows

For the square normalising flow, we consider the situation where $\mathcal{L} = L$ such that we remain on an equally fine lattice. Moreover, we take the map $\mathcal{T}_\theta$ to be bijective with the inverse $\mathcal{T}_\theta^{-1}$.

The transformation induces a change of variables under the statistical integral starting from the base distribution $p_L(\varphi)$, such that the density changes according to

$$\tilde{p}(\tilde{\phi}) = p_L(\varphi) |\det J_{\mathcal{T}}(\varphi)|^{-1} \,, \quad \text{with} \quad \varphi = \mathcal{T}_\theta^{-1}(\tilde{\phi}) \,. \tag{13}$$

Here, $J_{\mathcal{T}} = \partial\, \mathcal{T}_\theta(\varphi)/\partial\varphi$ denotes the Jacobian of the transformation $\mathcal{T}_\theta$. It is a square matrix since the transformation does not change the dimensionality of the fields. The (log-)likelihood computation must be tractable, which is one of the main challenges when constructing normalising flows. In particular, efficient evaluations of the $\log \det J_{\mathcal{T}}$ term, where $J_{\mathcal{T}}$ scales with the lattice volume, are imperative.

Square normalising flows are typically performant on relatively coarse lattices but scale poorly when increasing the number of sites [21–23]. Moreover, when simulating lattice field theories in the broken phase with its multi-modal distribution, some normalising flow models struggle to perform well due to mode collapse [20, 24, 25]. Lastly, in the vicinity of the phase transition, the critical slowing down problem is generally not solved but instead transferred into the learning process [21].

## 3.3 Rectangular normalising flows

In this paper, we aim to connect coarse and fine lattices with a normalising flow. This alleviates some aforementioned issues, as the fine configurations can inherit correlations from the used coarse samples. Moreover, autocorrelation times are typically under control in coarse theories, and efficient sampling is possible [5, 9].

Nevertheless, we cannot directly apply square normalising flows in these situations. To connect the coarse lattice $\Lambda_L$ with linear lattice size $L$ to the fine lattice $\Lambda_{\mathcal{L}}$ with linear lattice size $\mathcal{L}$, where $\mathcal{L} > L$, we require a map

$$\mathcal{T}_\theta : \ \mathbb{R}^{L^d} \to \mathbb{R}^{\mathcal{L}^d} \,, \tag{14}$$

which maps the coarse field $\varphi$ to the fine field $\tilde{\phi}$. However, the change of variables formula (13) is only applicable when the target and the base distribution relate to the same number of lattice sites.

Instead, we consider the injective map $\mathcal{T}_\theta$ for which the more general change of variables formula [17, 23, 26],

$$\tilde{p}(\tilde{\phi}) = p(\varphi) \left| \det J_{\mathcal{T}}^T(\varphi) J_{\mathcal{T}}(\varphi) \right|^{-1/2} \,, \tag{15}$$

with $\varphi = \mathcal{T}_\theta^\dagger(\tilde{\phi})$, is applicable. Here, $\mathcal{T}_\theta^\dagger$ denotes the left inverse of the map $\mathcal{T}_\theta$, such that

$$\mathcal{T}_\theta^\dagger \circ \mathcal{T}_\theta(\varphi) = \varphi \,. \tag{16}$$

Since we not only need the Jacobian determinant but also the matrix product of the now non-square Jacobians, the likelihood computation is even more demanding.

This concludes the recap on square and rectangular normalising flows and leaves us with the desired application depicted in Figure 1. We show the coarse lattice with configurations $\varphi$ distributed according to some not necessarily trivial distribution $p_L$ and the map $\mathcal{T}_\theta$ to the finer lattice configurations $\tilde{\phi}$ which approximately follow the target distribution $p_{\mathcal{L}}$.

Lastly, one can also see how the rectangular normalising flow relates to learning block spinning transformations and their inverses. When $\mathcal{T}_\theta^\dagger$ respects the symmetries of the lattice field theory and maps to all possible configurations $\varphi$, it constitutes a block spinning transformation. Then, the learned map $\mathcal{T}_\theta$ can, in the appropriate sense, be thought of as an inverse renormalisation group step.

In the following section, we will first show the proposed architecture concretely and then discuss the invertibility of the transformation $\mathcal{T}_\theta^\dagger$ further.



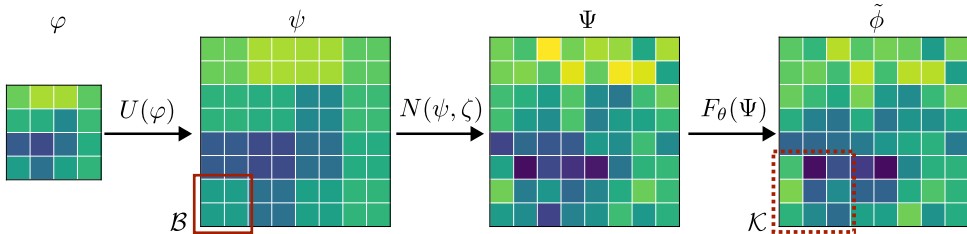

Figure 1: Visualization of the rectangular flow given by the optimisable transformations $\mathcal{T}_\theta : \mathbb{R}^{L^d} \to \mathbb{R}^{\mathcal{L}^d}$ and $\mathcal{T}_\theta^\dagger : \mathbb{R}^{\mathcal{L}^d} \to \mathbb{R}^{L^d}$. Here the coarse field $\varphi$ is distributed according to a Boltzmann distribution $p_L$ and the push forward $\tilde{\phi}$ is thought to be approximately distributed according to the target Boltzmann distribution $p_\mathcal{L}$.

# 4 Architecture

In this Section we discuss the details of constructing a normalising flow for the purpose of sampling a lattice field theory on a fine lattice starting from a coarse one. Our goal is to enable an efficient likelihood computation while maintaining a solid connection to the renormalisation group.

As was already mentioned above, evaluating the change of variables (15) proves difficult in this situation as we require $\det J_\mathcal{T}^T J_\mathcal{T}$ to be tractable. Many approaches use Hutchinson samples to estimate this Jacobian determinant [2,17,27]. However, in contradistinction to the above approaches, the target distribution in a lattice field theory is already known, but we lack the tools to sample from it efficiently. Additionally, we require the exact likelihood of each flowed sample to construct a Markov chain on the target distribution and obtain exact samples for the desired theory.

Here, we propose an architecture that instantiates the desired normalizing flow of Figure 1. It is close to the intuition of the renormalisation group and allows us to track the likelihood of the configurations throughout the flow precisely. For this, we only assume that we have sampling access to the theory on the coarse lattice and can compute the action on a fine lattice. The introduced transformation maps the coarse configurations to a finer lattice by which it enlarges the linear lattice size of the coarse lattice by a factor of $b = 2$ as shown in Figure 2.

Each building block will be explained in the following paragraphs. Moreover, we will show how the log-likelihood can be computed precisely at each step and comment on the notion of their invertibility.

Figure 2: Illustration of the architecture of the flow $\mathcal{T}_\theta$. Given a coarse configuration $\varphi \in \mathbb{R}^{L^d}$, the flow upsamples it naïvely ($U$), applies Gaussian noise in an invertible manner ($N$), and pushes the noised configuration through a continuous normalising flow ($F_\theta$) with optimisable parameters $\theta$. Here $\mathcal{B}$ denotes the naïve upsampling blocks over which also the invertible noise is applied. Moreover, $\mathcal{K}$ denotes the local kernel of the continuous normalising flow.

## 4.1 Naïve upsampling

To retain the correlations embedded in the coarse configurations and to ensure a tractable Jacobian determinant of the rectangular normalising flow, we naïvely upsample the coarse configurations, following the example in [28]. From the fields $\varphi_x$ on each coarse lattice site, we create a $b^d$ block $\mathcal{B}$ of sites $\psi_{x \in \mathcal{B}} = \varphi_x$ on the finer level. Having reshaped the data into a $L^d$ dimensional vector, the Jacobian of this transformation is given by

$$
J_U = \begin{pmatrix}
1 & 0 & \cdots & 0 \\
\vdots & 0 & \cdots & 0 \\
1 & 0 & \cdots & 0 \\
0 & 1 & \cdots & 0 \\
0 & \vdots & \cdots & 0 \\
0 & 1 & \cdots & 0 \\
\vdots & & & \vdots \\
0 & 0 & \cdots & 1 \\
0 & 0 & \cdots & 1
\end{pmatrix}_{(bL)^d \times L^d} .
\tag{17}
$$

The Jacobian determinant reduces to a constant which is straightforwardly computed as $\log |\det J_U^T J_U| = L^d \log(b^d)$. By this construction, we choose the left inverse $U^\dagger$ of the transformation $U$ to be given as the average over the blocks $\mathcal{B}$ used in the naïve upsampling. This relates the operation to a standard block spinning transformation. Accordingly, the new fields and their log-likelihood after the naïve upsampling are given by

$$
\psi = U(\varphi), \qquad \varphi = U^\dagger(\psi), \qquad \log p_U(\psi) = \log p_L(\varphi) - \frac{1}{2} L^d \log(b^d).
\tag{18}
$$

## 4.2 Adding invertible noise

In general many configurations on the fine lattice will lead to the same configuration on the coarse lattice. Moreover, starting from the coarse lattice, we have no information on the fine (UV) degrees of freedom. This motivates the addition of noise, $\zeta$. It ensures that the configuration $\tilde{\phi}$, we map to, is not solely determined by the coarse (IR) degrees of freedom, $\varphi$, but also acknowledges the missing information from the UV. Furthermore, inflating the configuration space with noise already finds many applications in the context of normalising flows [4, 6, 29, 30].

To allow for the invertibility of the transformation in the way explained in Section 4.4, we add the noise to each upsampling block $\mathcal{B}$ such that it sums up to zero. This is easily achieved by sampling the noise variables $\zeta_{\mathcal{B},i}$ (with $i = 1, \ldots, b^d - 1$) of each block from a multivariate Gaussian distribution with covariance matrix

$$
\Sigma = \sigma_\theta^2 \left( \mathbb{I}_{b^d - 1} - \frac{1}{b^d} \right),
\tag{19}
$$

where $\mathbb{I}_{b^d - 1}$ is the $b^d - 1$ dimensional identity matrix and $\sigma_\theta$ is a single learnable parameter. The final noise component is determined by the averaging constraint, such that

$$
\zeta_{\mathcal{B},b^d} = -\sum_{i=1}^{b^d - 1} \zeta_{\mathcal{B},i}.
\tag{20}
$$

Since the noise is sampled independently of $\psi$, the new fields and their log-likelihood are given by

$$
\Psi = \psi + \zeta, \qquad \varphi = U^\dagger(\Psi), \qquad \log p_N(\Psi) = \log p_U(\psi) + \log p_\zeta(\zeta).
\tag{21}
$$

Here, $p_\zeta$ denotes the distribution of the noise. Furthermore, $U^\dagger$ also constitutes the left inverse of the flow after the noise addition, $(N \circ U)^\dagger = U^\dagger$. This originates in the fact that the noise vanishes when taking the mean over the upsampling blocks $\mathcal{B}$,

Because of the noise addition via (21), the full transformation $\mathcal{T}_\theta$ now not only depends on the coarse configuration $\varphi$ but also on the noise $\zeta$. Note that the coarse degrees of freedom $\varphi$ stem from $\mathbb{R}^{L^d}$ and the noise degrees of freedom for $\zeta$ come from $\mathbb{R}^{\mathcal{L}^d - L^d}$, as we ensure that the mean of the noise over the blocks is zero. Accordingly, the embedding of the coarse lattice configuration on the finer lattice and the noise ensures that the configuration $\Psi$ spans the full $\mathbb{R}^{\mathcal{L}^d}$ space. In particular, they do not only span the subspace of configurations that are compatible with the coarse lattice after the upsampling.

## 4.3 Normalising flow

We optimise a square normalising flow on the finer lattice $\Lambda_\mathcal{L}$ for mapping this upsampled and noised configuration $\Psi \in \mathbb{R}^{\mathcal{L}^d}$ to a configuration approximately following a target distribution. A similar approach to first embed the coarse degrees of freedoms into a larger space and then apply a normalising flow can also be found in [30, 31].

For their recent success we choose continuous normalising flows (CNF) for this task. A CNF is defined by a neural ordinary differential equation (NODE) [32] of the form

$$\frac{d\Psi(t)_x}{dt} = G_\theta^{(\mathcal{K})}(\Psi(t), t)_x. \tag{22}$$

Here $G_\theta^{(\mathcal{K})}(\Psi(t), t)_x$ is a function with the kernel $\mathcal{K}$ of the field $\Psi(t)$ during the fictitious time $t$ with optimisable parameters $\theta$. The great benefit of CNFs relies on the fact that one can implement symmetries of the theories directly into the flow and has great flexibility in the choice of $G_\theta^{(\mathcal{K})}$. We align our construction of $G_\theta^{(\mathcal{K})}$ closely to the proposed architecture in [33] and parametrise $G_\theta^{(\mathcal{K})}$ as

$$G_\theta^{(\mathcal{K})}(\Psi(t), t)_x = \sum_{y,d,f} W_{xydf} K(t)_d H(\Psi_y(t))_f. \tag{23}$$

Here, $W$ denotes a learnable weight matrix, $K(t)$ a time kernel, and $H$ a basis expansion of the field $\Psi_y(t)$. To ensure the $\mathbb{Z}_2$ symmetry of the $\phi^4$-theory, we choose the expansion

$$H_1(x) = x, \qquad H_f(x) = \sin(\omega_f x), \tag{24}$$

with learnable frequencies $\omega_f$, $f = 2 \dots F$. For the time kernel $K(t)_d$ we choose the first $D$ terms of the Fourier expansion on the time interval $[t_1, t_2]$ we integrate over when solving the differential equation (22). $F$ and $D$ are hyperparameters of the CNF.

The lattice index $y$ only runs over the sites inside the kernel $\mathcal{K}$ (see Figure 2). This entails that the flow can only directly couple sites in the scope of the kernel instantaneously, which reduces the number of learnable parameters. We choose to do so since the coarse correlations should already be present in $\Psi$. Hence, the flow must only learn the short-scale correlations on the finer lattice.

As noted in [33], better training is achieved when we parametrise the learnable weight matrix $W$ as

$$W_{xydf} = \sum_{d'f'} \tilde{W}_{xyd'f'} W^K_{d'd} W^H_{f'f}, \tag{25}$$

where the indices $d', f'$ run from one up to the bond dimensions $F', D'$ in time and frequency space, respectively. $F'$ and $D'$ also are hyperparameters of the CNF.

Furthermore, we identify spatial bonds $(x, y)$ in $W$ that are equal under the lattice symmetry. This further reduces the number of learnable parameters and force the flow to respect symmetries present in the lattice field theory.

As we consider a $\phi^4$-theory on a periodic two-dimensional lattice, we ensure that we respect the symmetry under 90° rotations and mirror reflections. Moreover, because of the upsampling in $b^d$ blocks, the field $\Psi$ admits to a $b$-step translational symmetry.

At the beginning of the training the flow is initialised at the identity, which means that at first no short scale correlations are added to the field, and they enter over the training. In each training step, the NODE can then be solved via the adjoint sensitivity method [32,33] with the initial conditions $\Psi(t_0) = \Psi$, such that

$$\tilde{\phi} = F_\theta(\Psi) = \Psi + \int_{t_0}^{t_1} dt\, G_\theta^{(\mathcal{K})}(\Psi(t), t). \tag{26}$$

This is clearly invertible by changing the time direction of integration. Moreover, the log–likelihood of the final configurations can be computed by [32]

$$\frac{d \log p_F(\Psi)}{dt} = -\nabla_\Psi \cdot G_\theta^{(\mathcal{K})}(\Psi(t), t). \tag{27}$$

Accordingly, after the application of the flow the final log-likelihood is given by

$$\log p_F(\tilde{\phi}) = \log p_N(\Psi) - \int_{t_0}^{t_1} dt\, \nabla_\Psi \cdot G_\theta^{(\mathcal{K})}(\Psi(t), t). \tag{28}$$

In analogy to discrete normalising flows, we will denote all the contributions to the log-likelihood by the transformation $\mathcal{T}_\theta$ by '$\log \det J_\mathcal{T}(\varphi, \zeta)$'. This includes the noise distribution $p_\zeta$ from (21). Hence, the log-likelihood of the full transformation $\mathcal{T}_\theta$ is given by

$$\log \tilde{p}(\tilde{\phi}) = \log p_L(\varphi) - \log \det J_\mathcal{T}(\varphi, \zeta). \tag{29}$$

Here, $\varphi$ and $\zeta$ denote the used coarse and noise degrees of freedom, respectively. Moreover, we specified here the used left inverse $\mathcal{T}_\theta^\dagger(\tilde{\phi}) = U^\dagger \circ F_\theta^{-1}(\tilde{\phi})$. This concludes the direct discussion of each element of the architecture and their interplay. To summarise, the proposed architecture naïvely upsamples the coarse configurations, adds noise, and then flows on the finer lattice. For each step, the log-likelihood can be computed exactly and efficiently. Furthermore, $\mathcal{T}_\theta : (\varphi; \zeta) \mapsto \tilde{\phi}$ and $\mathcal{T}_\theta^\dagger : \tilde{\phi} \mapsto \varphi$ are both fixed simultaneously. As a last step, we now also discuss the notion of invertibility for the transformation $\mathcal{T}_\theta$.

## 4.4 Invertibility

By investigating the proposed architecture, we note that while $\mathcal{T}_\theta^\dagger$ maps to a lower dimensional space, given *any* configuration $\phi \in \mathbb{R}^{\mathcal{L}^d}$, the coarse as well as noise degrees of freedom are always implicitly determined via

$$\varphi(\phi) = \mathcal{T}_\theta^\dagger(\phi), \qquad \zeta(\phi) = \Psi(\phi) - \psi(\phi) = F_\theta^{-1}(\phi) - U(\mathcal{T}_\theta^\dagger(\phi)). \tag{30}$$

This is in analogy to the work in [6,30], where the coarse and noise degrees of freedom are explicitly separated.

It also nicely fits to the intuition that renormalisation group transformations are not invertible because we loose information about the UV degrees of freedom. Accordingly, this architecture manifests that the transformation would indeed be invertible (as similarly touched

upon in [6, 34]), if we would keep the UV degrees of freedom exactly. Here, in practice, we do not keep the noise degrees of freedom exactly but fix the distribution they stem from.

Accordingly, although we have defined a left inverse $\mathcal{T}_\theta^\dagger$ that parametrises a block spinning transformation, it does so in the context of a bijective map on the finer lattice. So, this architecture parametrises manifestly invertible transformations between $(\varphi, \zeta) \in \mathbb{R}^{L^d} \times \mathbb{R}^{(\mathcal{L}^d - L^d)}$ and $\phi \in \mathbb{R}^{\mathcal{L}^d}$. Given a configuration $\varphi$ on the coarse lattice, we know the subspace of $R^{\mathcal{L}^d}$ it can map to when embedded with the additional noise $\zeta$. Moreover, given any configuration $\phi$ on the fine lattice, we can uniquely determine a coarse and a noise configuration $\varphi, \zeta$, from which it fictitiously stemmed. This way, we paid tribute to the fact that the renormalisation group is a semigroup while maintaining the invertibility of the transformation. More details on the interpretation of $\mathcal{T}_\theta^\dagger$ as a block spinning transformation can be found in Appendix B.

This concludes the description of the proposed architecture. When discussing the optimisation criterion in the next section, we also consider the freedom of choosing the base and target action and related applications.

## 5 Optimization criterion

We have arrived at a normalising flow architecture that can relate coarse and fine lattices. Our focus is now on the distributions induced by the actions $S_L$ and $S_\mathcal{L}$ on the respective lattices, which shall represent different discretisations of (roughly) the same continuum theory. With this, we follow an intuition also mentioned in [35].

We aim at optimising the described transformation $\mathcal{T}_\theta$ and the action on the coarse or fine lattice such, that the acceptance rate of the Markov chain on the fine grid is maximised.

Concerning the couplings, this section does not further specify whether we want to optimise the couplings of the base or target distribution. Whereas in Section 6, we focus on the optimal couplings on the coarse lattice for a given sampling problem on a target lattice, in Appendix A, we also discuss how one can sensibly adapt the couplings of the target distribution.

To this end, we consider the reverse Kullback-Leibler divergence between the target and the push forward distribution, $p_\mathcal{L}$ and $\tilde{p}_\mathcal{L}$ respectively,

$$D_{KL}(\tilde{p}_\mathcal{L} \| p_\mathcal{L}) = \int D\tilde{\phi} \, \tilde{p}_\mathcal{L}(\tilde{\phi}) \log \frac{\tilde{p}_\mathcal{L}(\tilde{\phi})}{p_\mathcal{L}(\tilde{\phi})} = D_{KL}^{(I)}(\tilde{p}_\mathcal{L} \| p_\mathcal{L}) + D_{KL}^{(II)}(\tilde{p}_\mathcal{L} \| p_\mathcal{L}). \tag{31}$$

We have split the expression in (31) into two terms: the first term holds the part of the divergence that relates to an expectation value over the push forward distribution,

$$D_{KL}^{(I)}(\tilde{p}_\mathcal{L} \| p_\mathcal{L}) := \mathop{\mathbb{E}}_{\tilde{\phi} \sim \tilde{p}_\mathcal{L}} \left[ S_\mathcal{L}(\tilde{\phi}) - S_L(\varphi(\tilde{\phi})) - \log \det J_\mathcal{T}(\varphi(\tilde{\phi}), \zeta(\tilde{\phi})) \right]. \tag{32}$$

The second term holds the part of the divergence that relates to the partition sums

$$D_{KL}^{(II)}(\tilde{p}_\mathcal{L} \| p_\mathcal{L}) := \log Z_\mathcal{L} - \log Z_L. \tag{33}$$

For most applications, one effectively drops the $D_{KL}^{(II)}$ term in (31): it is constant and irrelevant to the optimisation as long as the normalizations for the base and target distributions remain fixed. However, we want to tune the base or target distribution of the flow, making an optimisation of the $D_{KL}^{(II)}$ term necessary.

After all, in lattice field theory, we think of the lattice action $S[c](\varphi)$ with bare couplings $c$ and the field $\varphi$ as a discretised version of the continuum action $\mathcal{S}[\mathcal{C}][\phi]$ with the couplings $\mathcal{C}$ and the continuum field $\phi(x)$. The continuum action is approached by the lattice action in

the continuum limit and the lattice action is structurally given in the same way for any lattice $\Lambda$ and couplings $c$, see (3).

This leads us to the following consideration. Our flow connects two fields on different lattices $\Lambda_L$ and $\Lambda_{\mathcal{L}}$. Therefore, it is only reasonable to wish for the distributions on the two lattices to be induced by different discretisations $S_L[c_L](\varphi)$ and $S_{\mathcal{L}}[c_{\mathcal{L}}](\phi)$ given by structurally the same lattice action formula, for instance (3).

We know from the renormalisation group, that the lattice couplings change under this flow in a non-trivial way. Therefore, we either introduce the couplings of the base or target actions as learnable parameters.

While the full partition sum is intractable for training, practically, we are only interested in its gradient w.r.t. the training parameters (i.e. the couplings).

Expressing the dependence of the partition sum on the learnable couplings $c_\theta$ as $Z[c_\theta]$, we can calculate the necessary gradient. Here we use insights from the work on restricted Boltzmann machines and contrastive divergence [36],

$$\nabla_\theta \log Z[c_\theta] = \frac{1}{Z[c_\theta]} \nabla_\theta Z[c_\theta] = \frac{1}{Z[c_\theta]} \nabla_\theta \int D\phi\, e^{-S[c_\theta](\phi)} = - \mathop{\mathbb{E}}_{\phi \sim p[c_\theta]} [\nabla_\theta S[c_\theta](\phi)]\,. \quad (34)$$

The remaining gradient with respect to the learnable parameters is readily computed. For most actions of interest, the number of couplings is small. Then, the expectation values can be reliably estimated using Monte Carlo techniques that require only a modest number of samples, rendering the direct computation of the gradient affordable during training.

In the $\phi^4$-theory investigated here, the gradients are given by

$$\frac{\partial S}{\partial \kappa_\theta} = -2 \sum_{\substack{x \in \Lambda \\ \mu}} \phi_x \phi_{x+\mu}\,, \qquad \frac{\partial S}{\partial \lambda_\theta} = \sum_{x \in \Lambda} \left( \phi_x^4 - 2\phi_x^2 \right)\,. \quad (35)$$

With this, we are not only able to tune the transformation between the coarse and fine action but also the couplings of the actions themselves. Furthermore, one can also consider other, more general, actions on the coarse lattice that allows for terms not present in the target fine action. This is left to future work. Here, we also note that the resultant couplings optimize the sampling quality on the fine lattice and must not follow the line of constant physics.

During training we will use two sets of Monte Carlo samples: one set which we push through the network to compute the gradient of $D_{KL}^{(I)}$. The other set is never pushed through the network but is used to compute the gradient of $D_{KL}^{(II)}$ via (34).

Here, we can already note that the couplings, for which we compute the $D_{KL}^{(II)}$ gradient, do not change too much between each training step. Moreover, since the respective set of samples is never pushed through the normalising flow, we can safely reuse them. Accordingly, one can use reweighing techniques to significantly reduce the number of required Monte Carlo simulations during training. For a given set of Monte Carlo samples with couplings $c_\theta$, the $\log Z$ gradient with couplings $c_\theta'$ at a later time in training can be estimated by

$$\nabla_\theta \log Z[c_\theta'] = - \frac{\mathop{\mathbb{E}}_{\phi \sim p[c_\theta]} \left[ \left( \nabla_\theta S[c_\theta'] \right) \frac{e^{-S[c_\theta'](\phi)}}{e^{-S[c_\theta](\phi)}} \right]}{\mathop{\mathbb{E}}_{\phi \sim p[c_\theta]} \left[ \frac{e^{-S[c_\theta'](\phi)}}{e^{-S[c_\theta](\phi)}} \right]}\,. \quad (36)$$

The above procedure is applicable if the distributions $p[c_\theta]$ and $p[c_\theta']$ are sufficiently close to each other. Since the couplings do not change too much during training and eventually converge at the end of it, the overlap between the distributions $p[c_\theta]$ and $p[c_\theta']$ at the two training times is typically quite large.

This concludes our discussion of the optimisation criterion.

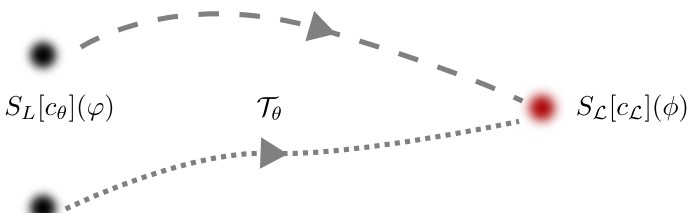

Figure 3: Illustration of the IR-Matching. The field $\varphi$ on the coarse lattice with action $S_L[c_\theta]$ and optimisable couplings $c_\theta$ is connected to the field $\phi$ on the fine lattice with action $S_\mathcal{L}[c_\mathcal{L}]$ and fixed couplings $c_\mathcal{L}$ by the transformation $\mathcal{T}_\theta$.

# 6 IR-matching

In this Section we discuss an application of the proposed architecture and optimisation criterion, which we coin *IR-Matching* and is illustrated in Figure 3. Concretely, we fix the action $S_\mathcal{L}[c_\mathcal{L}]$ on the fine lattice with couplings $c_\mathcal{L}$ and find the optimal transformation $\mathcal{T}_\theta$ and coarse couplings $c_\theta$ needed to sample from the target distribution.

First we will discuss more details of the approach. Then its applicability in different phases for differently fine target lattices is considered. Details on the used hyperparameters and error estimation can be found in Appendix C.

## 6.1 Details of IR-matching

At the beginning of training, the couplings of both lattices have the same value. Then, during training, the chosen couplings on the coarse lattice move to the values optimal for the sampling problem at hand.

As the coarse couplings enter the sampling process of the coarse configurations $\varphi$, we must use a sampling algorithm on the coarse lattice that is differentiable w.r.t the learnable couplings. Here, we use a standard batched Langevin sampling algorithm, where the fields follow the Langevin dynamics in the fictitious time $\vartheta$ according to

$$\varphi^{(\vartheta+1)} = \varphi^{(\vartheta)} - \tau \nabla_\varphi S[c_\theta](\varphi^{(\vartheta)}) + \sqrt{2\tau}\, \eta(\vartheta), \tag{37}$$

where $\eta(\vartheta)$ denotes a Gaussian noise term and $\tau$ the Langevin time step. We thermalise the set of coarse lattice configurations once and only re-thermalise for the next training step with a slightly different set of couplings. We note in this context that there are more sophisticated methods that also utilise the adjoint method for the Langevin stochastic differential equation [37, 38]. However, we found it sufficient for our purposes to simply backpropagate through the Langevin process. One of the reasons is that the lattices we sample from are quite small, and hence the memory cost required for the backpropagation was not too demanding.

To sample on the fine lattice, we draw coarse configurations $\varphi$ as described above and apply transformations $\mathcal{T}_{\theta_i}$ with $i = 1, \dots, \log_b(\mathcal{L}/L)$ with different weights $\theta_i$ successively. This yields

$$\tilde{\phi} = \mathcal{T}_\theta(\varphi) = \mathcal{T}_{\theta_{\log_b(\mathcal{L}/L)}} \circ \dots \circ \mathcal{T}_{\theta_1}(\varphi), \tag{38}$$

belonging to the fine lattice $\Lambda_\mathcal{L}$. Each application of $\mathcal{T}_{\theta_i}$ thereby doubles the linear lattice size of the lattice. The samples $\tilde{\phi}$ on the fine lattice are then used to compute the $D_{KL}^{(I)}$ gradient of the loss from (32). Notably, the $D_{KL}^{(II)}$ gradient from (34) only requires Monte Carlo samples on the coarse lattice $\Lambda_L$, rendering this gradient estimation computationally efficient.

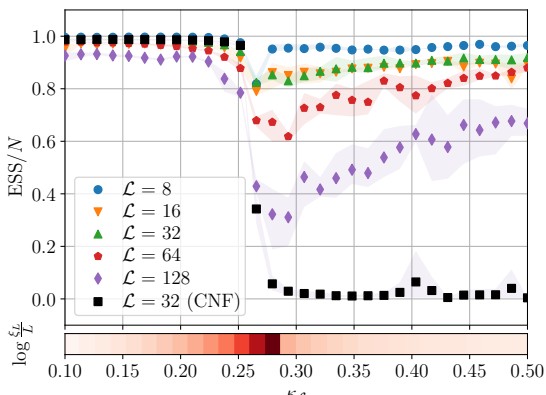

(a) ESS/$N$ for the IR-Matching method on differently fine lattices and a CNF with $\mathcal{L} = 32$ for comparison.

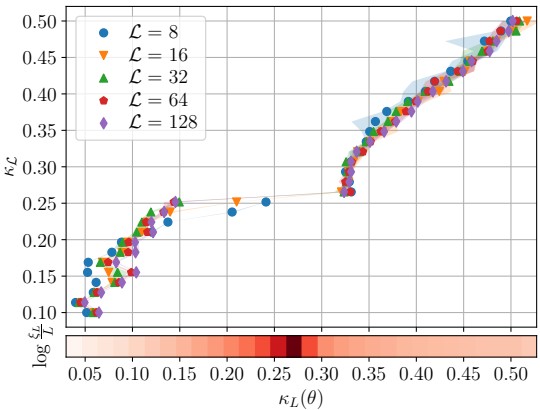

(b) Optimizable coupling $\kappa_L(\theta)$ on the coarse lattice relative to the fixed couplings $\kappa_{\mathcal{L}}$ on the fine lattices.

Figure 4: Results of the IR-Matching method for different fine lattice sizes $\mathcal{L}$ starting from a coarse lattice with $L = 4$. The couplings on the fine lattice were fixed and only $\kappa_L(\theta)$ on the coarse lattice was optimisable. The bottom colorbars indicate the transition between phases by showing the $\log \xi_L / L$ for the x-axis' couplings.

## 6.2 Results

To put the IR-Matching method to work, we test its performance across the phase diagram of a two-dimensional $\phi^4$-theory.

For each normalising flow, the coarsest grid is given by a two-dimensional lattice with a linear lattice size of $L = 4$. To gain a better intuition for the evolution of the couplings, we only leave one coupling open for optimisation. Specifically, we fix the on-site coupling $\lambda = 0.01$ and only optimise the hopping-term $\kappa_L(\theta)$ for each targeted lattice size and fine coupling. Nevertheless, optimising multiple couplings is entirely feasible and unproblematic.

We also want to give an intuition for the phase transition and the related correlations. Therefore, we show the logarithm of the correlation length $\xi_L$ relative to the linear lattice size $L$ of the coarse lattice as a function of $\kappa_{\mathcal{L}}$ in the colorbar at the bottom Figure 4a and Figure 4b. The system is in the symmetric phase for small values of $\kappa$ and enters the broken phase when increasing its value.

In Figure 4a, we show the ESS/$N$ after training the models to saturation. The shaded areas indicate the error of the ESS/$N$. As the latter is especially noisy in the broken phase, the errors are more fluctuant here. To begin with, the IR-Matching method works very well in the symmetric phase, even for lattices as fine as $128 \times 128$, obtaining an ESS/$N$ of well over 80%.

Near the critical region, we expect to see remnants of critical slowing down, since it mirrors the physics of the system. And, indeed we notice a drop in performance. Nevertheless, especially for lattices up to $64 \times 64$ the ESS/$N$ remains high enough not to harm practical use, steadily achieving ESS/$N$ over 60%.

Moreover, even in the broken phase where normalising flows often fail due to mode collapse, the ESS/$N$ rarely drops below 40% and for most lattices remains around or over 80%. The ability of the flow to also work well in the broken phase, characterised by a multi-modal weight, is easily understood, as our base distribution is already multi-modal.

For comparison, we also plot the ESS/$N$ for a standard CNF for the $32 \times 32$ lattice as described in Section 4.3 but now with full translational symmetry and a kernel that spans the entire system. We see that it achieves comparable performance in the symmetric phase but is not performant when entering the broken phase. We train the CNF as described in

Appendix C, but adopt the hyperparameters ($F = 30, D = 10, F' = 20, D' = 20$) using the notation established in Section 4.3 and use a learning rate of $5 \times 10^{-3}$ for stability. Notably, we significantly reduce the number of learnable parameters. This originates in the fact that we can heavily restrict the CNF kernel for the IR-Matching. In this case, the standard CNF for the $32 \times 32$ lattice has approximately twice as many learnable parameters as the IR-Matching method. Moreover, for the next larger $64 \times 64$ lattice, the standard CNF has approximately five times as many learnable parameters.

Since the actions on the coarse and fine lattice roughly correspond to different discretisations of the same continuum theory, we can relate well the couplings $\kappa_L(\theta)$ and $\kappa_{\mathcal{L}}$ on both levels. This is shown in Figure 4b.

We display the coarse lattice couplings $\kappa_L(\theta)$ on the x-axis together with $\log \xi_L/L$ to indicate the phase transition, while the y-axis shows the fixed couplings $\kappa_{\mathcal{L}}$. The optimisable coarse couplings cleanly sever the plot into two regions. These regions are related to the deeply symmetric and broken phase. A gap in $\kappa_{\mathcal{L}}$ emerges close to the (smeared out) phase transition. This means that for simulating physics close to the transition on the finer lattice, the optimal coarse-level couplings lie in the symmetric or broken phase but not close to the critical physics. This behaviour is an advantage of the IR-Matching method, as simulating the symmetric or broken phase on the coarse lattice is easier than simulating in the critical region.

Relating the couplings is especially intuitive when we think of $\kappa_L(\theta)$ as the lattice coupling after some coarse-graining steps starting from the finer lattice at coupling $\kappa_{\mathcal{L}}$. The critical point of the theory is an unstable fixed point with respect the to block spinning transformation $\mathcal{T}_\theta^\dagger$ in coupling space. Accordingly, applying successive RG steps drives the bare couplings away from the critical point when starting in its vicinity on the fine level.

To further stress the practical benefit of the IR-Matching method, we show the training times for each lattice in Figure 5. The used hardware is described in Appendix C. We trained flows for different lattice sizes up to $\mathcal{L} = 128$, again starting from a coarse lattice with $L = 4$. For the bare couplings, we use $\kappa_{\mathcal{L}} = 0.25$ and $\lambda_{\mathcal{L}} = 0.01$, approaching the critical region from the symmetric phase. The bold lines show the moving average of the ESS/$N$, whereas the shaded lines indicate its momentary value.

We directly observe that the lattices with $\mathcal{L} \leq 32$ are trained readily in under one hour and also systems like the $64 \times 64$ lattice are trained in under seven hours. For the $128 \times 128$ lattices, we generally see convergence of the ESS/$N$ after around 48 hours of training.

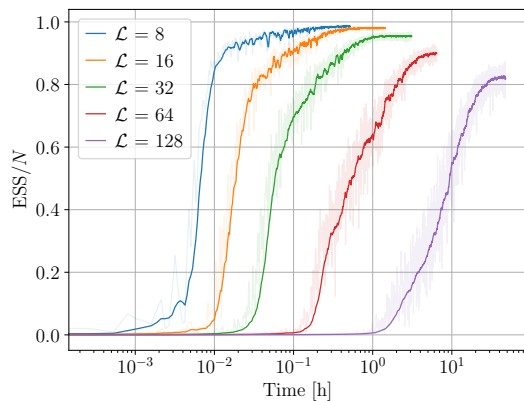

Figure 5: ESS/$N$ for the IR-Matching method for different linear lattice sizes $\mathcal{L}$ of the target lattice. The bare coupling $\lambda_{\mathcal{L}} = 0.01, \kappa_{\mathcal{L}} = 0.25$ are used for all finer lattices, which brings the system into the vicinity of the critical region from the symmetric phase. The bold lines show the moving average whereas the shaded lines show the ESS/$N$ during training.

Since the IR-Matching introduces the coarse lattice as an optimisable object, its training times and performance may be significantly improved by using finer lattices to start with, offering a structural way to improve the flow, going beyond making the learned map more expressive.

# 7 Conclusion

In this paper, we have developed normalising flows for lattice field theories, inspired by the renormalisation group. We have proposed an architecture, that is well suited to find transformations between a coarse and a fine lattice with respect to a common action formula. Moreover, we also also introduced the *IR-Matching* method, as a practical application, see Section 6. To our knowledge this novel method outperforms current benchmarks for normalising flows regarding the applicability in the broken phase and the extension to $128 \times 128$ systems.

In the IR-Matching method from Section 6, we fix the couplings on the finer lattice and optimise the couplings on the coarse lattice. Furthermore, in Appendix A, we also introduce the *UV-Matching* method. There we we fix the couplings on the coarse lattice and aim to find an iterable transformation to finer lattices, where the fine lattice couplings become optimisable parameters.

In the application of the IR-Matching method we showed that one can efficiently sample a $\phi^4$-theory in two dimensions on lattices as large as $128 \times 128$ while only requiring Monte Carlo configurations from a $4 \times 4$ system. Moreover, we found that the couplings are driven away from the critical region when coarsening. Conversely, this allows us to use samples from the symmetric or broken phase on the coarse lattice to sample from the critical region on the fine lattice. This is indeed expected from the renormalisation group and further benefits the efficiency of the sampling process. Moreover, the training times for the IR-Matching method are very reasonable, as for instance the $64 \times 64$ systems were trained in only a few hours.

Furthermore, just as for standard normalising flows, this approach instantiates a rather general sampling technique for a lattice field theory, only requiring plugging in the action and drift of the theory. Accordingly, the proposed method is straightforwardly generalised to any other scalar models. Applications to long-ranged and fermionic theories [39] are direct avenues for future research. Moreover, since the sampling process naturally moves through different resolutions, it potentially ties in nicely with multi-level Monte Carlo approaches on the lattice [40].

At last, we have discussed the capacity of our approach to combine the benefits of traditional MCMC methods and normalising flows. However, since normalising flows typically perform well on coarse lattices, one could first train a standard coupling-conditional normalising flow on a very coarse discretization. Then, this renormalisation group inspired flow may be used to upscale the generated configuration, leading to a normalising flow all the way down. This is left to future work.

By incorporating the coarse action into the optimisation process, along with the linear size of the coarsest lattice, we have introduced a practical approach that extends beyond making the machine learning model more expressive. This opens up many new interesting and physics-informed avenues for normalising flows in lattice field theory and beyond. Specifically, we aim at a combination of the current framework with the physics-informed renormalisation group [41] and the flow-based density of states [42]. We hope to report on these combinations in the near future.

# Acknowledgments

We thank Friederike Ihssen, Julian Urban, Lingxiao Wang, and Kai Zhou for discussions and collaborations on related subjects.

**Funding information**   This work is funded by the Deutsche Forschungsgemeinschaft (DFG, German Research Foundation) under Germany's Excellence Strategy EXC 2181/1 - 390900948 (the Heidelberg STRUCTURES Excellence Cluster) and the Collaborative Research Centre SFB 1225 (ISOQUANT). We thank ECT* for support at the Workshop "Machine Learning and the Renormalization Group" during which this work has been developed further. The authors acknowledge support by the state of Baden-Württemberg through bwHPC.

# A   UV-matching

In Section 6, we have introduced the IR-Matching method. In this approach we kept the fine lattice couplings $c_{\mathcal{L}}$ constant and optimised the coarse lattice couplings $c_L(\theta)$ as well as the transformation $\mathcal{T}_\theta$. For the *UV-Matching* as illustrated in Figure 6, we are interested in the opposite case: we keep the coarse lattice couplings constant and optimise the transformation $\mathcal{T}_\theta$ as well as the fine lattice couplings $c_{\mathcal{L}}(\theta)$.

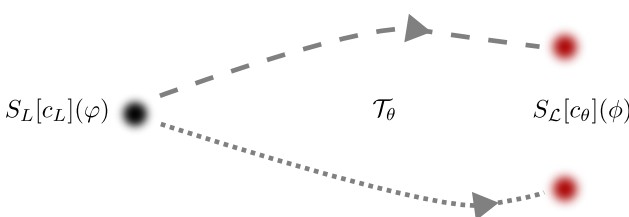

Figure 6: Illustration of the UV-Matching. The field $\varphi$ on the coarse lattice with action $S_L[c_L]$ and fixed couplings $c_L$ is connected to the field $\phi$ on the fine lattice with action $S_{\mathcal{L}}[c_\theta]$ and optimisable couplings $c_\theta$ by the transformation $\mathcal{T}_\theta$.

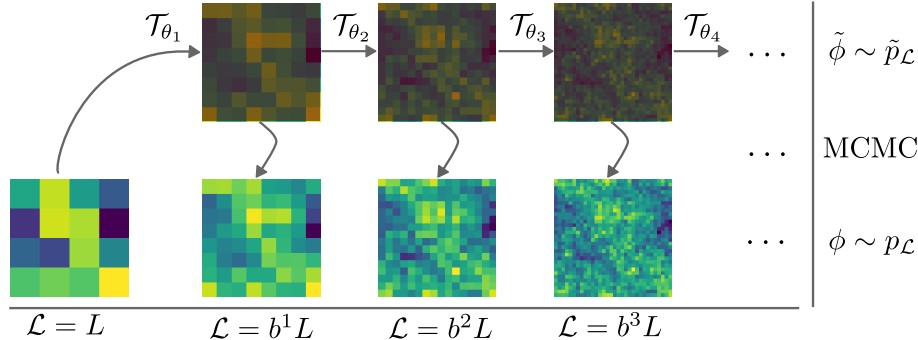

Figure 7: Iterative application of the learned flow $\mathcal{T}_{\theta_i}$ with learnable parameters $\theta_i$. After each application of the transformation one receives samples $\tilde{\phi} \sim \tilde{p}_{\mathcal{L}}$ with $\mathcal{L} = b^i L$. Then, a MCMC accept reject step from Equation (10) can be conducted to ensure that one samples from the $\phi^4$-theory distribution $p_{\mathcal{L}}$ on the finer lattice.

## A.1 The method in detail

To begin with, the general training procedure is very similar to the one described for the IR-Matching. We sample coarse configurations $\varphi$, push them through the transformation $\mathcal{T}_\theta$. Then we minimise the Kullback-Leibler divergence from (31), but in contradistinction to the IR-Matching method we leave the fine lattice couplings open for optimisation. Accordingly, we do not just optimise the overlap between the target and push forward distribution by changing the latter. Instead we allow for the target distribution to change slightly in the direction of the current push forward.

As the coarse lattice couplings are now fixed, we can use a traditional HMC algorithm to draw samples from the coarse distribution. However, optimising the couplings on the fine lattice requires us to draw configurations of the respective system during training to estimate the gradient from (34).

This has the potential to become computationally expensive and to re-introduce critical slowing down. It is here, where the reweighing approach from (36) turns out to be very useful. Since the configurations used to compute the $\log Z$ gradient are not pushed through the network, we can safely reuse them. By using reweighing the need to draw samples from the target distribution at every learning step is reduced, significantly lowering the number of Monte Carlo samples required for training. Note that reweighing is only possible if the coupling changes slowly. One could in principle also use configurations from the machine learning model during retraining steps to estimate the $\log Z$ gradient. However, this is left to future work.

Notably, the learned transformation $\mathcal{T}_\theta$ does not explicitly depend on the linear lattice size. This is straight–forwardly true for the naïve upsampling. Furthermore, the system size independence also applies for the rest of the architecture in Figure 2, as the addition of noise is conducted per upsampling block $\mathcal{B}$ and the normalising flow learns a kernel $\mathcal{K}$ of fixed size across the lattice.

Accordingly, one can learn the transformation $\mathcal{T}_\theta$ on a coarse lattice where training and sampling is efficient. Then one applies this transformation iteratively many times to finer lattices. This iterative application is illustrated in Figure 7, where we start from a coarse lattice with linear lattice size $L$ and double the linear lattice size after each application, $\mathcal{L} = b^i L$.

In spirit this is very similar and indeed inspired by the works in [5, 28], where the authors use convolutional neural networks. The great benefit of our approach is that we can efficiently keep track of the log–likelihoods of the samples. Firstly, this enables the direct application of Monte Carlo methods to make the approach exact, see Figure 7. Secondly, it also allows us to directly monitor the efficiency of the method, which is indicated by the acceptance rate. This allows one to make an educated assertion of the maximal number of applications of the transformation.

## A.2 Results

The training only directly determines the optimal coupling after the first iteration of the transformation $\mathcal{T}_{\theta_1}$, where we double the linear lattice size of the coarsest field. As we want to apply the transformation iteratively, we require a sensible approach to choose the optimal coupling after further iterations. Here, we straight–forwardly choose the couplings that maximise the $\text{ESS}/N$ on the respective lattice.

Showing the estimation of the coupling explicitly is especially illustrative in the one-dimensional case: each iteration only doubles but not quadruples the total number of lattice sites and most changes vary less strongly than in higher dimensions. We thus match a one-dimensional $\phi^4$-theory on a lattice with linear lattice size $L = 8$ and couplings $\kappa_L = 0.1, \lambda_L = 0.01$ to a lattice with linear lattice size $\mathcal{L} = 16$, fixed coupling $\lambda_\mathcal{L} = \lambda_L$, and optimisable coupling $\kappa_\mathcal{L}(\theta)$.

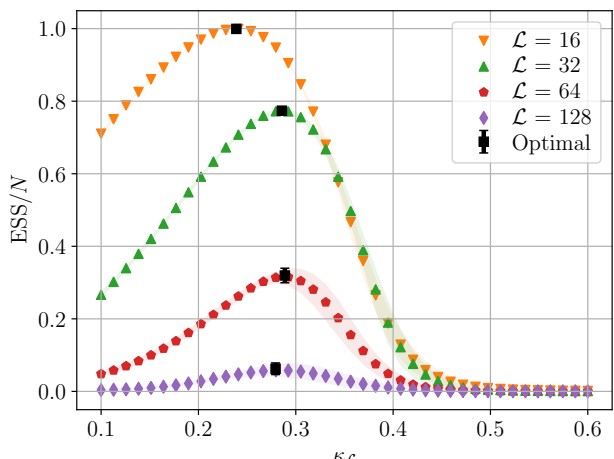

Figure 8: Estimation of the coupling $\kappa_{\mathcal{L}}(\theta)$ for a one-dimensional $\phi^4$-theory by optimising the ESS/$N$ after iterative application of the UV-Matching flow.

In Figure 8, we show the ESS/$N$ for different couplings after iterative application of the flow. We firstly note that the ESS/$N$ varies smoothly, making it easy to estimate the optimal value. While we see that the straight–forward iteration of the learned flow works, it is not apt to reach finer lattices efficiently as the ESS/$N$ drops approximately exponentially.

To improve performance, we note that the straight-forwardly iterated transformation may not be optimal, but should also not be too far away from the optimal transformation. Accordingly, only a few retraining steps $N_{RT}$ can suffice to obtain good results. For retraining, we again sample with the flow on the respective fine lattice we are currently optimising and minimise the KL divergence from (31) with a learning rate that we reduced by a factor of five. Other than that everything else stays the same.

In Figure 9, we provide the optimal ESS/$N$ for the learned flow after applying $N_{RT}$ retraining steps. As can be seen, we reach good performance up to lattices with $\mathcal{L} = 512$ for the one-dimensional $\phi^4$-theory and the ESS/$N$ saturates after already 50 retraining steps. In practice, we use early–stopping, checking whether the ESS has saturated or not, to determine the optimal number of retraining steps.

We now examine the behaviour of the UV–Matching method across different phases on lattices of varying sizes, and discuss how it is related to the physics of the system. Here, we again return to the two-dimensional lattice. Specifically, we fix coarse lattices in both the symmetric and broken phases near the critical region, and aim to optimise the transformations $\mathcal{T}_{\theta_i}$.

In Figure 10a, we plot the ESS/$N$ after each iteration of the flow to lattices with a linear size $\mathcal{L}$. The starting couplings are taken from the coarse lattice, as shown on the y-axis, along with the respective values of $\log \xi_L / L$ to provide context for the critical region. First, we observe that the ESS/$N$ remains high in almost every case for lattices with sizes up to $\mathcal{L} = 32$. Even on the $64 \times 64$ lattice, the ESS/$N$ remains above 50% in both the deeply symmetric and broken phases. However, starting from the critical region on the coarsest lattice, we notice a cone of decreasing ESS/$N$ when moving to finer grids.

To make sense out of this, we show the learned couplings relative to the couplings on the coarsest lattice in Figure 10b. In contrast to the IR-Matching, the couplings actually move towards their critical values when applying the inverse block spinning transformation iteratively in the vicinity of a phase transition. This originates in the fact that we now optimise the couplings on the finer lattice. Accordingly, as sampling and training on the critical point posits a computationally challenging problem, the ESS/$N$ drops significantly in each iteration.

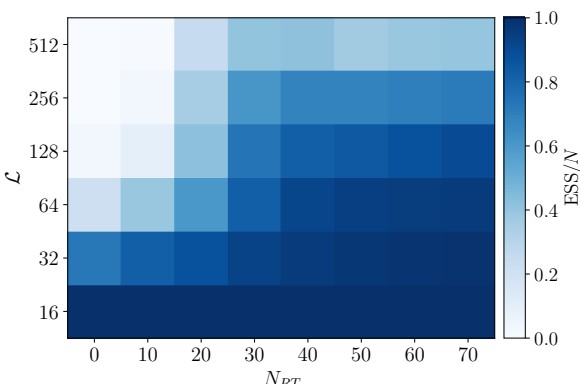

Figure 9: The optimal ESS/$N$ for the learned flow from Figure 8 after $N_{RT}$ retraining steps.

When comparing the behaviour of the couplings for the IR- and UV-Matching in Figure 4b and Figure 10b respectively, we see the plateaus for the critical coupling $\kappa_{\mathcal{L}}$. For the IR-Matching method, this plateau manifests as a gap, where the coarse lattice couplings move away from the plateau. For the UV-Matching method, the couplings actually move towards it.

Accordingly, while the UV-Matching method should not be understood as a method to simulate the whole parameter set of the field theory, it lends itself to simulating critical physics on the lattice.

## B  Relation to block spinning transformations

In the main text, we introduced renormalisation group inspired normalising flows. The connection between the actions on the coarse and fine lattice $(S_L, S_{\mathcal{L}})$ can be made explicit when we assume that the action on the fine lattice is given.

In the following, we will call a transformation $\mathcal{T}_\theta^\dagger : \mathbb{R}^{\mathcal{L}^d} \to \mathbb{R}^{L^d}$ a *block spinning transformation* when it does not violate any symmetries of the original lattice and for the induced blocking.

$$P(\varphi|\phi) := \delta\left[\varphi - \mathcal{T}^\dagger(\phi)\right], \tag{B.1}$$

it is true that

$$P(\varphi|\phi) \geq 0, \quad \forall \phi, \varphi, \qquad \int \mathcal{D}\varphi \, P(\varphi|\phi) = 1, \quad \forall \phi. \tag{B.2}$$

Then, when $\phi$ follows the above-mentioned Boltzmann distribution, the block spinning transformation also induces a Boltzmann distribution for the coarser field $\varphi$ via

$$e^{-S_L(\varphi)} = \int D\phi \, P(\varphi|\phi) \, e^{-S_{\mathcal{L}}(\phi)}. \tag{B.3}$$

This construction then ensures that the transformation leaves the partition sum invariant, as is directly seen by

$$Z_L = \int D\varphi \, e^{-S_L(\varphi)} \stackrel{\text{(B.3)}}{=} \int D\varphi \, D\phi \, P(\varphi, \phi) \, e^{-S_{\mathcal{L}}(\phi)} \stackrel{\text{(B.2)}}{=} \int D\phi \, e^{-S_{\mathcal{L}}(\phi)} = Z_{\mathcal{L}}. \tag{B.4}$$

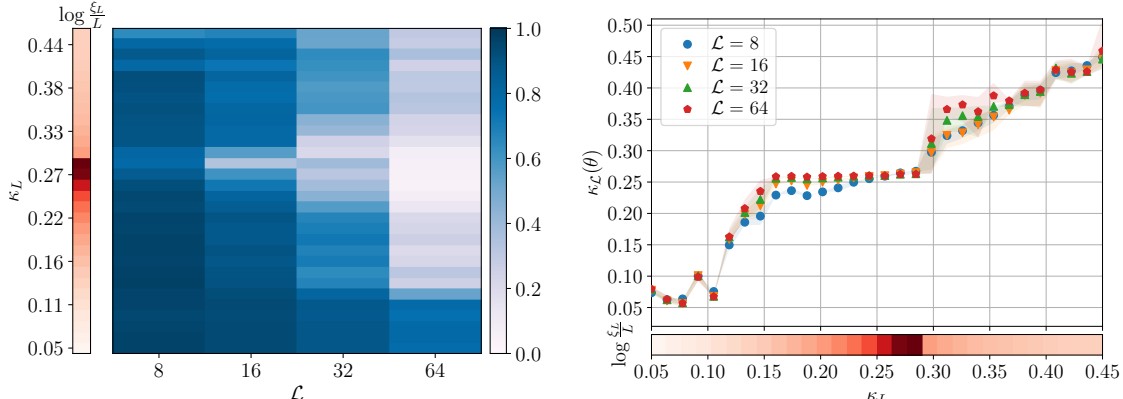

(a) ESS/$N$ for the UV–Matching for different linear lattice sizes $\mathcal{L}$ across the phase structure of the system.

(b) Optimizable coupling $\kappa_{\mathcal{L}}(\theta)$ for the UV–Matching.

Figure 10: Results for the UV-Matching method for different number of iterations of the transformation $\mathcal{T}_\theta$ starting from a coarse lattice with $L = 4$. The couplings on the coarse lattice were fixed and only $\kappa_{\mathcal{L}}(\theta)$ on the fine lattice was optimisable. To indicate the transition between phases, we display colorbars for $\log \xi_L/L$ on the coarse lattice.

Given the context of (B.3), we now want to consider how the actions on the coarse and fine lattice are connected to each other when we consider $\mathcal{T}_\theta^\dagger$ as a block spinning transformation. Starting from the fine action and applying (B.3) directly leads us to

$$e^{-S_L(\varphi)} = \int D\phi \, \delta\left[\varphi - \mathcal{T}_\theta^\dagger(\phi)\right] e^{-S_{\mathcal{L}}(\phi)}. \tag{B.5}$$

Now, we would like to connect the fine field $\phi \in \mathbb{R}^{\mathcal{L}^d}$ to the coarse field $\varphi \in \mathbb{R}^{L^d}$ in a more explicit manner. To this end, we revert the steps indicated in Figure 2 and, hence, apply the normalising flow $F_\theta$ inversely. Under the integral, we now call $\Psi = F_\theta^{-1}(\phi)$ and using (26) together with (B.5), we obtain

$$e^{-S_L(\varphi)} = \int D\Psi \, \delta\left[\varphi - \mathcal{T}_\theta^\dagger \circ F_\theta(\Psi)\right] e^{-S_{\mathcal{L}}(\phi) + \log \det J_F(\Psi)}. \tag{B.6}$$

Next, we note that we can express each variable $\Psi$ according to (21) and (18) as $\Psi = U(\varphi') + \zeta$. More precisely, we now only consider the noise degrees of freedom $\zeta' \in \mathbb{R}^{\mathcal{L}^d - L^d}$ and can connect them to the full noise field via (20). We do so because now we may define the field $\psi' = (\varphi', \zeta')^T$ that has the same dimension as $\Psi$, and we can rewrite it as

$$\Psi = U_\psi(\psi') = U(\varphi') + V(\zeta'). \tag{B.7}$$

We can equally rewrite the integration over $\Psi$ as an integration over $\psi'$ or $\varphi'$ and $\zeta'$ respectively. The respective Jacobian $J_{U_\psi}$ of this transformation is field-independent as $U$ and $V$ are linear transformations relating to (18) and (20) respectively. So, this transformation only picks up a constant factor. Accordingly, we obtain from (B.6) and (B.7)

$$e^{-S_L(\varphi)} \simeq \int D\varphi' D\zeta' \, \delta\left[\varphi - \mathcal{T}_\theta^\dagger \circ F_\theta \circ U_\psi(\psi')\right] e^{-S_{\mathcal{L}}(\phi) + \log \det J_F(\Psi)}. \tag{B.8}$$

Now, we can see that $F_\theta \circ U_\psi$ is precisely the transformation $\mathcal{T}_\theta$ for a fixed noise field described in Figure 2, leading to the further simplification

$$e^{-S_L(\varphi)} \simeq \int D\varphi' D\zeta' \, \delta\left[\varphi - \mathcal{T}_\theta^\dagger \circ \mathcal{T}_\theta(\varphi'; \zeta')\right] e^{-S_\mathcal{L}(\phi) + \log \det J_F(\Psi)} . \tag{B.9}$$

Accordingly, we can use the left inverse property of the transformation $\mathcal{T}_\theta$ from (16), such that $\mathcal{T}_\theta^\dagger \circ \mathcal{T}_\theta(\varphi'; \zeta') = \varphi'$. This enables us to perform the integration over $\varphi'$, and we obtain from (B.9)

$$e^{-S_L(\varphi)} \simeq \int D\zeta' \, e^{-S_\mathcal{L}(\phi) + \log \det J_F(\Psi)} . \tag{B.10}$$

As in the equations before, we have kept the implicit notation, such that $\phi = \mathcal{T}_\theta(\varphi; \zeta')$ and $\Psi = U(\varphi) + V(\zeta')$ under the integral. The result from (B.10) shows nicely, that the block spinning transformation as implemented by $\mathcal{T}_\theta^\dagger$ literally boils down to an integration over the noise degrees of freedom. Moreover, it lets us explicitly connect the coarse action $S_L$ to the fine action $S_\mathcal{L}$.

## C Hyperparameters and error analysis

In this Appendix we describe the hyperparameters used for our architecture.

For the naïve upsampling, there are no hyperparameters present. Furthermore, for the addition of noise, at the beginning of the training, we initialise the learnable variance $\sigma_\theta^2$ of the Gaussian noise $\zeta$ to equal the variance of the configurations $\varphi$ on the coarse lattice. This is done to ensure that the noise does not dominate the flow at the beginning of the training.

To compute the $\log Z$ gradients, we sample $3k$ configurations on the respective coarse lattice with a standard Hybrid Monte Carlo (HMC) algorithm. For thermalisation, we start from an i.i.d initial Gaussian configuration and run the Hybrid Monte Carlo Code until $2k$ proposals have been accepted. We save a sample after each HMC step.

Since reweighing is used in this part of training, we have to choose a point at which we want to sample new configurations. For this, we also store the likelihood of the configurations when they are sampled. The ESS/$N$ for each new coupling is then computed, and we choose a lower bound $R_W$ for the ESS/$N$ upon which we sample anew. In our experience, $R_W = 0.85$ gives a good compromise between speed and accuracy.

For the IR-Matching from Section 6, we use a batched Langevin Sampling algorithm from (37) on the coarsest lattice. We use a step size of $\tau = 0.01$ for this. Training starts from an i.i.d. initial Gaussian configurations and involves thermalising the system for $5 \times 10^4$ steps. Between each training step, we re-thermalise the system by taking 500 Langevin steps, starting from the current batch of configurations.

For the CNF, we achieved good results when using $F = 11$ learnable sine frequencies and choosing $D = 10$ for the kernel number. The time and frequency space bond dimensions are $F' = 20$ and $D' = 20$, respectively. In all computations, the coarsest lattice of the flow has a linear lattice size of $L = 4$ to showcase that one only needs very coarse lattices. Hence, the kernel size is also limited. We found a kernel size of 2 sufficient for the IR-Matching as presented here and used a kernel size of 3 for the UV-Matching.

For optimisation, we use the Adam optimiser with decaying parameters $\beta_1 = 0.8$ and $\beta_2 = 0.9$ as suggested in [33]. The initial learning rate is set to 0.01 and decays exponentially after each learning step by a factor of 0.997. In all cases, we used a batch size of 256 as done in [33].

To estimate the errors for the ESS/$N$ and the optimised couplings, we trained each model three times with different random seeds. The data point is then given by the respective mean value computed from the respective saturated regions of the training. The error bars of the ESS/$N$ reach from the maximally to the minimally obtained value.

For the training and evaluation we used the resources of the bwHPC cluster. Virtually all the trainings were conducted on a system with an NVIDIA Tesla V100 GPU with 32 GB of memory, supported by Intel Xeon Gold 6230 processors (2 sockets, 40 cores total), and 384 GB of main memory. For the larger 128× 128 lattice, we switched to a system with an NVIDIA A100 GPU with 80 GB of memory, supported by Intel Xeon Platinum 8358 processors (2 sockets, 64 cores total), and 512 GB of main memory. The latter switch was done to conveniently accommodate the memory requirements of the larger lattice sizes.

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
