# Peer review of "Super-Resolving Normalising Flows for Lattice Field Theories"

_SciPost Physics, doi:SciPost Phys. 19, 077 (2025)_

## Round 1 · Referee Report · Anonymous (Referee 1) · 2025-4-24

Report

Super-Resolving Normalising Flows for Lattice Field Theories

Marc Bauer, Renzo Kapust, Jan M. Pawlowski, Finn L. Temmen

The paper explores new ways to merge normalising flow, a machine learning method, with Markov chain Monte Carlo techniques in lattice field theory. The paper is interesting and very well written, and deserves to be published after addressing the issues below.

1) My main comment is very general. The following questions kept bothering me, until they were partly settled in Section V: how are the theories on the fine and the course lattice related, given that they are supposed to describe the same 'physics'; since lattice couplings depend on the ultraviolet cutoff scale, how is their relation ('flow') established? Do the transformations as the lattice volume increases keep the theory on a line of constant physics (LCP)?

This is partly addressed in Section V, but ideally the conceptual idea should be presented earlier.

1a) Let me go into detail now. The authors state that the physical size of the lattice is kept fixed and that the lattice spacing 'a' is reduced under their transformation. Couplings in the lattice action (i.e. at the UV scale) depend on the cutoff scale. In Section V, this is addressed, studying the flow of couplings, denoted as c[a]. It is then made clear that the theory on the coarse lattice is not known a priori, but has to be learned from the constructed theory on the fine lattice. This is a nontrivial conceptual step, which as I stated should be mentioned earlier. It is also slightly unsatisfactory, since it means we need to learn both the transformation moving to finer lattices, and the flow of couplings moving to coarser lattices. A pessimist might say that hence we know nothing on either side! An optimist would say that this problem is now solved. What do the authors say?

1b) What can be addressed better is whether during the entire scaling up of the lattice volume, from L = 4, 8, 16, 32, 64, 128, the theory is moving on a line of constant physics. This I could not deduce from the paper, i.e. need the couplings on the coarse lattice be redetermined for every volume, or is there indeed a trajectory in coupling space? If they need to be redetermined, does this make the calculation more and more expensive going to larger lattice volumes? It would be elegant to stay on the LCP.

2a) It is stated that to start with a lattice of size L=4 helps in building in correlations. I wonder how useful this is close to the critical point. On a 4x4 lattice the transition is hardly visible in e.g. the magnetic susceptibility. The critical coupling depends on the lattice size, $\kappa_c(L)$, which can be studied using finite size scaling. Hence close to the critical point, the correlations on 4x4 may be not representative to those on 128x128, and even in the 'wrong' phase, due to the L dependence of the critical coupling. As I understand from Fig 3, the transformation is learned from an ensemble which is not related to the ensemble on the fine lattice via the RG flow. I wonder whether the authors have studied (or can study) the dependence on the initial ensemble on the coarse lattice.

2b) Potentially related to this is fig 4b and the gap in the $\kappa_L$ (horizontal) direction. The gap seems to reflect the uncertainty of where the transition sits on the coarsest lattice. Does the gap gets smaller when starting from 8x8 instead of 4x4?

3) I appreciate the connection with the inverse RG, especially ref [5], and the discussion in App A. In ref [5] the flow is exactly in the opposite direction compared to in the body of the paper, and each (inverse) RG step takes one closer to the critical point, as mentioned by the authors. I wonder whether both approaches could be combined. Have the authors considered this?

4) Very briefly, $\sigma_\theta^2$ in eq 19 is just one parameter, or does it depend on steps in the algorithm?

Recommendation

Ask for minor revision

  • validity: -
  • significance: -
  • originality: -
  • clarity: -
  • formatting: -
  • grammar: -

Author:  Renzo Kapust  on 2025-07-04  [id 5616]

(in reply to Report 1 on 2025-04-24)

We thank the referee for their insightful comments. We have changed the manuscript according to their remarks and now want to take the time to answer the remaining questions according to the numbering from the referee report.

  1. We agree with the referee that the relation between the theories on the coarse and fine lattice should be mentioned earlier and we changed the manuscript accordingly (see revised pages 1 and 8).

Concerning the line of constant physics: Our approach is inspired by the renormalisation group and uses mechanisms related to it. In the manuscript, we wanted to make these nice conceptual connections visible as we believe that they are also important for the further understanding and development of normalising flows. In the end, the primary goal of this approach is, however, a technical one: We want to enable normalising flows to perform better on fine lattices and (relatedly) close to criticality. We achieve this by choosing a non-trivial distribution on the coarse lattice and by constructing an RG-informed transformation towards the fine lattice. During the development, we indeed checked that the transformation thereby does $not$ generally move along the line of constant physics. Nevertheless, constructing flows that do so is still an interesting avenue for future research in our group.

1.a We thank the referee for pointing out this crucial step in our construction, namely that for super-resolving normalising flows, one also learns the couplings of the coarse lattice theory. While we understand that this can be perceived as a drawback as it requires one to learn the couplings $\textit{additonally}$ to the flow, we fully take the optimist standpoint in this regard.

Indeed, it is a non-trival question whether the optimization of the coarse couplings is feasible in practice. However, this is precisely what we have shown in this manuscript. With this, we included a further optimizable object, the coarse base distribution, into the normalising flow framework. As we are only interested in the correct (fine) target lattice distribution, we should also use the freedom we have when choosing the (coarse) base distribution. A good guess for the coarse base distribution is structurally the same as the fine target distribution. However, one can also include further operators on the coarse lattice, which may improve training.

We want to stress here, that the action on the fine lattice is fixed, so we exactly know the target on one side of the flow. For the rest of the flow, we allow all the freedom we have, to optimally sample from the desired distribution.

1.b We agree with the referee that it would be elegant to move on the line of constant physics. Here, we take the more general approach and focused on optimal performance w.r.t. the fine target distribution. We also now mention explicitly that we do not generally move along the line of constant physics (see revised page 8). The couplings should be optimized for each volume seperately. This we now also state in the manuscript explicitly (see revised page 9). While one could also learn a function for the evolution of the couplings, we did not experiment with this. However, it should be stated that the only conceptually necessary part of the coarse coupling optimization is the computation of the expectation values

$\underset{\varphi \sim p_L[c_\theta]}{\mathbb{E}}[\partial_{c_\theta}\;Sc_\theta] $

on the coarse lattice (Equation 34 in the manuscript) which has a constant computational cost independent of the chosen fine lattice $\Lambda_\mathcal{L}$.

2.a We thank the referee for this question as it revolves around many important aspects of this work. To begin with, it is true that we expect the flow to benefit from the correlations of the coarse theory. In comparison to widely used non-interacting base distributions, choosing an interacting coarse base distribution is one way to include more correlations already in the prior. So, comparatively speaking, we may still benefit from the used prior. However, as was mentioned in the question, it is also true that there are more and less beneficial couplings to include these wanted correlations in the base distribution. After all, this is why we optimize the coarse coupling.

Furthermore, it is true that the initial coupling plays a role when it comes to the practical question of how well the optimization of the coarse coupling works. This one can indeed probe experimentally by training flows with different initial couplings. However, since we just introduced the super-resolving normalising flows here, we did not yet perform a systematic study w.r.t. all initial parameters of the model. Instead, we found that choosing the initial coarse couplings to be the same as the fine couplings results in a good performance. Of course when one trained one super-resolving normalising flow for some coupling and lattice size, one can use the learned coarse coupling as an initial guess for the training of further models.

2.b Yes, we have run simulations starting from a $8 \times 8$ lattice. Then the plateau becomes significantly smaller and the transition towards the plateau is more gradual. Importantly, the effect is preserved that the coarse couplings move away from the critical point.

3. Indeed, this work was partly inspired by ref [5]. Since super-resolving normalising flows directly allow to compute the log-likelihood of the samples in a tractable way, one could indeed proceed with the same approach as in ref [5] and replace the convolutional networks with the architecture put forward in this manuscript. This would directly allow for reweighting approaches or the inference of the emerging operators mentioned in the outlook of ref [5].

4. Indeed, for each upsampling layer, $\sigma_\theta^2$ is just one parameter. It is initialised to be proportional to the variance of the field values from the coarse base distribution. We now stress this explicitly in the manuscript (see revised page 6).

---

## Round 1 · Referee Report · Anonymous (Referee 2) · 2025-5-2

Report

This work is a study of normalizing flows for lattice field theories, a class of ML methods explored in recent years to improve lattice Monte Carlo calculations. In particular, this work explores the use of "super-resolving" normalizing flow architectures to sample lattice scalar field theories, which employ an RG-inspired hierarchical structure. The work also proposes and studies a new avenue for improving model quality, optimizing the parameters of the base distribution alongside the flow model parameters, called "IR matching". They find that better performance can be obtained than with an unstructured continuous normalizing flow (CNF) architecture, especially in the symmetry-broken phase of the scalar field theory.

This is a nice piece of work overall. The statistical formalism is correct and (other than some notational issues and missing details) the presentation is clear and readable. The numerical results appear to be thorough and demonstrate that the proposed approach is effective for the problem treated. Absent a few details, the descriptions should be sufficient to reproduce the results. The paper goes out of its way to make physical interpretations of various features of the ML setup and results, which is interesting and informative.

There are two notable but excusable weaknesses to the paper: 1) the application to scalar field theory, and 2) the emphasis on the "super-resolving" structure as a novel aspect of this work. On 2), RG-inspired hierarchical architectures have been explored and discussed since the earliest studies of this subject: the Neural Network Renormalization Group (NNRG) paper was one of the very first flows for lattice papers. On 1), scalar field theory is a common testbed for these methods and excellent results have been obtained for it using many different flow model approaches; it is not a challenging theory to model, or sample from in the first place. Nevertheless, given the expense of testing in more complicated systems like non-Abelian gauge theories, it's still worth first checking whether an idea works in scalar theory. The most convincing demonstrations in scalar field theory are of methods that generalize without complications to gauge theories, but that is not the case for the hierarchical architectures discussed here. The generalization of this sort of approach to gauge theories is complicated, as explored in Ref [8].

However, the "IR matching" aspect of this work is novel so far as I know, generalizes immediately without any complications to any lattice field theory including QCD, and seems like a promising avenue for exploration to construct improved flow models. I don't think it's necessary to restructure the paper around this point (although I would not object if the authors wanted to do so), but I do think some adjustment of the framing and discussion of novelty would be good. This aspect of the work by itself is more than sufficient to meet the journal's acceptance criteria, once minor issues are addressed.

Some detailed comments and questions for the authors:

  1. As noted above, the paper needs some explicit discussion added of what are the novel aspects versus previous works. For example, it seems that the super-resolving architecture proposed here, where noise is fed in in hierarchical upscaling steps, is the same as in NNRG. Other than fine details, it seems that the new parts over NNRG are the use of an ODE flow, as well as the "IR Matching" part of the training procedure.

  2. What batch size is used for training? What batch size is used in ESS evaluations? (The ESS can be extremely inflated at small batch sizes, so this is important.) These should be noted somewhere.

  3. The ESS for a fixed standard flow (when it can be evaluated on multiple volumes) scales between volumes exponentially like $ESS(V) \approx ESS(V_0)^{V/V_0}$. For a plot like Fig. 5 with a standard flow, using this relation to rescale and compare all the curves at some fiducial volume $V_*$ will result in curve collapse (up to finite-volume effects at small volumes and effects due to volume averaging $\approx$ batch size). However, for these super-resolving flows, there is extra structure relating different volumes, which might change the picture. If you can think of some fair comparison, it would be very interesting to know whether volume scaling properties are qualitatively different for these super-resolving flows.

  4. Some comments are made about avoidance of mode collapse in the symmetry-broken phase, but as written, these conclusions seem to be based on the numerical value of the ESS. When evaluated on insufficient samples, the ESS can be artificially inflated when mode collapse occurs (it can appear to be finite when it should be $\approx 0$). Are you explicitly verifying that the model distribution is bimodal?

  5. The left inverse of the naive upsampling isn't unique, block spinning with any weights that sum to one will invert the operation just as well. Why doesn't this cause ambiguities in which Jacobian needs to be computed?

  6. Equation 20 introduces some additional breaking of translational symmetry: the noise added to the first $b^{d-1}$ sites is uncorrelated site-to-site, but the noise added to the last site is correlated with everything else. Is there anywhere else in the architecture that can learn to compensate for this, or is it necessarily inherited by the final model density?

  7. On page 8, it says $D^{(II)}_{KL}$ can be estimated using only Monte Carlo samples from the coarse lattice. Why is this so, if $\log Z_L$ and $\log Z_\mathcal{L}$ both appear in Eq. 33?

  8. Notational quibble: the flow map is only a diffeomorphism when considered as a flow between (coarse dof, noise dof) and (fine dof). As defined in Eq.~8, the super-resolving flow map $\mathcal{T}_\theta$ is not a diffeomorphism, so in Eq. 9, $\tilde{p}(\tilde{\phi})$ is not the push-forward of a density on the coarse degrees of freedom $\varphi$, it is a pushforward of a density over $(\varphi, \zeta)$. Similarly, $\log \det J_\mathcal{T}(\varphi)$ in Eqs. 29 and 32 should really be $\log \det J_\mathcal{T}(\varphi, \zeta)$. In a related point, Eq. 15 is technically correct as written, but it's maybe worth noting that because $\mathcal{T}^\dagger_\theta$ is many-to-one, $\tilde{p}{\tilde{\phi})$ is a constant over all $\tilde{\phi}$ that map to the same $\varphi$, and these constants tile the fine-grid manifold $\tilde{\phi}$.

Recommendation

Ask for minor revision

  • validity: high
  • significance: ok
  • originality: ok
  • clarity: high
  • formatting: perfect
  • grammar: excellent

Author:  Renzo Kapust  on 2025-07-04  [id 5617]

(in reply to Report 2 on 2025-05-02)

We thank the referee for their detailled and insightful comments as we think that they improved the quality of the manuscript. We changed the latter in the respective ways and now want to answer the remaining questions according to the numbering from the referee report.

1.
Indeed, we perceive the main novel contributions to be the IR-Matching aspect of the super-resolving normalising flows. With respect to the upsampling and noise addition contribution, we already referenced Ref [6,28,29,30] at the respective places to show the connection to existing work. Of course, we already cited Ref [4], the NNRG, which we perceive to be a great contribution to the topic. Now, we also added the reference to Ref [4] at the same place and want to highlight the explicit comment made in the text towards the existing literature (see revised page 5). A related comment is already present below Eq. 30. We now also added a comment to highlight the focus on the IR-Matching aspect of this paper (see revised page 1).

2.
We thank the referee for this comment, since the batch size should of course be included in the manuscript. Similar to Ref [33], we used a batch size of 256 for training the flows. The values for the ESS stated in Fig. 4.a and 10.a, were computed by averaging three independently trained flows with the same parameters but different random seeds. Each model was trained well into a region where the loss and ESS saturated. The ESS for that model was then determined by averaging the ESS in this saturated region. Exemplarily, we also tested this method by computing the ESS for a trained model by taking $\sim 10^5$ samples. We state this now also explicitly in the manuscript (see revised page 15).

3.
This was indeed also a question we asked ourselves. While we did not test it exhaustively for the IR-Matching, we did check it more thorougly for the UV-Matching in App. A. Here, we trained a super-resolving normalising flow, and then copied the weights of one full upsampling layer to a new one. Then, one still observes an approximately exponential decay of the ESS when iterating this procedure. However, the weights are already well initialised, meaning that only a few training steps are necessary to reach a good performance.

4.
Indeed, we also checked for the avoidance of mode collapse by considering the distribution of individual lattice sites exemplarily for all lattice sizes in the broken phase. Moreover, as seen in Fig. 4.a, we also show the ESS of a standard CNF, where the ESS was determined in the same way as for the other models. Here you see that the ESS of the super-resolving normalising flows is qualitatively better than the one of the standard CNF. Accordingly, it would be surprising if the good ESS for the super-resolving normalising flows is purely a side effect of the used sample size, as one would expect the same effect for the standard CNF.

5.
As for instance used in Refs [15, 20, 21], the likelihood of the upsampled samples does only depend on the specific form of the left inverse in the following way (see Eq. 15 in the manuscript):

$p(\psi) = p(\;U^\dagger(\psi)\;) \left| \det\left[\; J_U^T(\;U^\dagger(\psi)\;) \; J_U(\;U^\dagger(\psi)\;)\;\right]\right|^{-1/2}\;,$

where in this context $U$ is the naive upsamling operator from the paper and $U^\dagger$ is some left inverse of it as described in the referee report. The variable $\psi$ denotes an upsampled lattice configuration that now lives on the finer lattice. Notably, the naive upsampling from the manuscript is a linear operator, meaning that the field-dependence from $J_U(\;U^\dagger(\psi)\;)$ drops out either way. More than that, $U^\dagger$ in the above equation only serves the purpose of mapping the upsampled configuration $\psi$ back to original coarse lattice configuration, meaning that its spefic form or Jacobian never enters the likelihood computation.

6.
We thank the referee for this comment. Indeed, the addition of the noise variables introduces an additional breaking of the translational symmetry. This breaking is something the model has to (and potentially can) unlearn. More elaborate ways to sample the noise degrees of freedom that do not break the translational symmetry may therefore improve the quality of the model.

7.
Indeed, for the IR-Matching, the computation of the $D_{KL}^{(II)}$ term only requires samples on the coarse lattice. This is because for the IR-Matching, only the couplings on the coarse lattice are optimized and the targeted distribution is fixed. Accordingly, the partition sum $Z_\mathcal{L}$ is a constant with respect to the learnable parameters. This is different for the UV-Matching as discussed in App. A.

8.
We thank the referee for their comments on the notations. We changed the notations in the manuscript that were pointed out (see revised pages 3, 6, 7).

---

## Round 2 · Referee Report · Anonymous (Referee 1) · 2025-7-16

Strengths

1- novel approach to combine normalising flow with RG inspired ideas in lattice field theory 2- clearly written 3- provides ample scope for further exploration

Weaknesses

1- the RG aspect is only “RG inspired”, one could quibble about what a proper RG transformation would do (but we won’t)

Report

The paper is clearly written and accessible. The authors have incorporated the referees' comments. The paper is relevant to those doing research in lattice field theory as well as in machine learning interested in this area.

Requested changes

No further changes requested.

Recommendation

Publish (easily meets expectations and criteria for this Journal; among top 50%)

---

## Round 2 · Referee Report · Anonymous (Referee 2) · 2025-8-27

Report

I wish to thank the referees for addressing my comments and questions. All of my substantive outstanding concerns have been alleviated. I have one last minor issue to point out, as well as a few parting comments and optional minor suggestions that the authors may choose to incorporate. However, these points are not critical, so I am happy to recommend the work for publication in any case without further review.

  1. It seems like a term for the density of the noise draws $\zeta$ is missing from the loss as defined in equations 31-33. It should be present formally (unless it is implicit in some of the other definitions). More importantly, the noise distribution has a learnable parameter $\sigma_\zeta$, which must enter somewhere in the loss in order to be optimized.

Follow-ups on previous points:

  1. The authors are correct that there are no obvious indications of any problem with the ESS evaluation at batch size 256.

There is a separate but related point that is of minor importance in the context of the present work, but which is worth taking into consideration in the future. In the paper, the ESS estimates and uncertainties are computed from the ESSes of 3 different models, obtained by training identically up to random seed. Although this not an unreasonable thing to do when some notion of uncertainty is desired, I would argue there are better and safer ones. - Averaging ESSes over models quantifies what typical sampler quality is achieved by some training protocol of interest, but one is not obligated to consider or sample multiple different models. Instead, one can and should simply take the most performant model found during training. The typical model quality isn't interesting, only the best achievable one. - When an error on the ESS is desired, it is better to evaluate it using multiple independent batches for a fixed model. Taking an average over multiple models mixes together the finite-batch uncertainty on the ESS with unrelated training noise. - It is more natural to average the ESS in inverse (or more generally consider the statistics of 1/ESS). This also avoids missing when the ESS is truly near-zero but high-variance in such a way that it typically evaluates as finite. This is an unfortunately common failure mode, but would be visible from the min/max-based definition of error bars used in this work, so it does not seem to be an issue here.

  1. Just to be clear, my point was that any violation of the symmetries of the true distribution in the model will result in reduced ESS. If the ODE flow kernel is translationally equivariant, it cannot possibly learn to correct for the breaking of translation symmetry by the noise insertion step. Using a non-equivariant ODE kernel can actually improve performance in this case by allowing it to compensate. However, this is a matter of optimization and not critical given the good performance already demonstrated.

  2. Thank you for addressing my quibble. However, one last bit of notation still merits quibbling over. The upscaling map $\mathcal{T}_\theta(\varphi)$, at least when the symbol is used as in Eq. 9, has an implicit $\zeta$ argument and should really be written $\mathcal{T}_\theta(\varphi,\zeta)$. As written, it makes it appear as if the flow is either injective into the fine space from the coarse space, rather than bijective from the coarse + noise spaces, or as if its definition involves a marginalization of some sort. I was significantly confused about this initially, so making it explicit could improve readability. However, one can figure out from the broader context that there must be an implicit $\zeta$ input, so this is not an essential change.

Recommendation

Publish (easily meets expectations and criteria for this Journal; among top 50%)

---

## Round 2 · Author Response

We thank the referees for their thoughtful comments as they helped to improve the quality of the paper. In this resubmitted manuscript, we clarified the role of the coupling optimization earlier in the text, corrected some notational quibbles, and clarified further points mentioned in the referee reports.

---

## Round 2 · List of Changes

• (Page 1) We now mention already in the introduction that the actions on the coarse and fine lattice have the same structural form. On the same note, we mention that the fine couplings are fixed and the coarse couplings are optimized w.r.t. the sampling quality on the fine lattice. This information would other wise be only provided in section V.
  • (Page3) In Eq. 9 we now refrain from calling $\Tilde{p}(\Tilde{\phi})$ the density's push forward, and merely introduce the notation where the $\Tilde{}$ referes to objects that were pushed through the map $\mathcal{T}_\theta$.
  • (Page5) We now stress that the noise scale is a single parameter for each upsampling layer $\mathcal{T}_{\theta_i}$
  • (Page 6) We now corrected a notational quibble. Because the learned map is only a diffeomorphism w.r.t (coarse dof, noise dof) and the (fine dof), we now also denote the dependence of the $\log \det J_\mathcal{T}$ term respectively.
  • (Page 7) Same as above.
  • (Page 8) We note that the couplings must not evolve along the lines of constant physics, but rather evolve during the training such that the sampling on the fine lattice becomes optimal.
  • (Page 9) We now mention explicitly that we train a flow (and correspondingly a coupling) for each fine targeted lattice size and fine coupling.

---

## Editorial Decision

published